# Influence of Sexual Dimorphism, Aging, and Differential Cell Capture Efficiency of Blood Separation Systems on the Quality of Platelet-Rich Plasma

**DOI:** 10.3390/jcm11061683

**Published:** 2022-03-18

**Authors:** Bibiana Trevissón, Ricardo Becerro-de-Bengoa-Vallejo, David Sevillano, Natalia González, Marta Elena Losa-Iglesias, Daniel López-López, Luis Alou

**Affiliations:** 1SALBIS Research Group, Nursing Department, Faculty of Health Sciences, Universidad de León, 24004 León, Spain; btrer@unileon.es; 2Facultad de Enfermería, Fisioterapia y Podología, Universidad Complutense de Madrid, 28040 Madrid, Spain; ribebeva@enf.ucm.es; 3Microbiology Section, Medicine Department, School of Medicine, Universidad Complutense de Madrid, 28040 Madrid, Spain; natgonzalez@med.ucm.es (N.G.); luisalou@ucm.es (L.A.); 4Faculty of Health Sciences, Universidad Rey Juan Carlos, 28032 Madrid, Spain; marta.losa@urjc.es; 5Research, Health and Podiatry Group, Department of Health Sciences, Faculty of Nursing and Podiatry, Industrial Campus of Ferrol, Universidade da Coruña, 15403 Ferrol, Spain; daniel.lopez.lopez@udc.es

**Keywords:** platelet-rich plasma, leukocytes, sexual dimorphism, aging, blood separation–concentration system

## Abstract

Few studies have checked the impact of the hormonal/immunological dimorphism of patients on the cellular composition of platelet-rich plasma products (PRP). Whole blood (WB) from 26 volunteers was concentrated using a device previously characterized. Platelet and white blood cell (WBC) counts in WB and PRP were compared between genders, and after the population was split into pre (≤50 years)- and post (>50 years)-menopausal ages. In WB, platelet–total WBC densities were comparable in men and women. The phagocytic cell composition differed (*p* ≤ 0.04). After dividing by ages, phagocytic cell discrepancies were linked to women > 50 years (*p* ≤ 0.05), and differences emerged in lymphocyte counts (lower in >50 years groups, within and between genders, *p* ≤ 0.05). In PRP, densities were significantly higher, but the PRP/WB ratios varied according to blood cell (lower for phagocytic cells) and between subjects (more favorable at a lower density of a particular blood cell in WB). This “system compensatory efficiency” reduced/reverted PRP differences in the leukocyte composition between genders/age–sex groups in WB. In PRP, neutrophils were higher in younger men than older women (*p* < 0.05). WB lymphocyte differences between age–sex groups persisted. Age is a more determining factor than sex in the preparation of PRP. Post-menopause, sexual dimorphism strongly influences the composition of leukocytes, also conditioned by the capture efficiency of the system.

## 1. Introduction

Autologous platelet-rich plasma (PRP) is a safe and effective blood therapy [1] that has emerged as a promising approach in regenerative medicine to repair and rejuvenate tissues damaged by injury or chronic diseases [2]. PRP therapy has gained popularity in the last two decades in several medical modalities, including sports medicine, orthopedic surgery, general surgery, dental and maxillofacial restorative surgery, and esthetics [1,3,4,5,6,7,8]. More recently, it has been successfully introduced as an adjunctive therapy in the treatment of diabetic foot ulcers (DFUs), encouraging further investigations in this field [9,10,11].

PRP is defined as the volume of the autologous plasma containing a platelet concentration above that of the whole blood (WB), but there is no clear specification of the required densities for effectiveness, which varies according to studies from >200 to 1000 × 10^6^ cell/mL [12,13]. PRP is obtained by separating concentrated layers of blood cells by centrifugation of WB and, depending on the method, made up of a variable concentration of platelets, WBCs, and hemodynamically active proteins [14,15,16].

The regenerative benefits of PRP are derived from the tandem action of the growth factors and various neurotransmitters and calcium released by platelets [1,17,18]. The co-infiltration of leukocytes provides antimicrobial and immunomodulatory properties and improves the proteomic content of PRP [19,20].

Currently, studies that provide strong evidence for the efficacy of PRP treatment are still needed, since serious inconsistencies from recent PRP clinical trials put into question the efficacy of thrombocyte therapy. Strong evidence suggests that PRP does not improve or has no additional benefit in plantar fasciitis, musculoskeletal soft tissue injuries, Achilles tendinopathy, or muscle strain [1,21,22,23,24]. Additionally, there is weak evidence for the benefit of PRP in tendon and ligament healing and knee arthritis [25,26].

By way of justification, these studies emphasize that clinical inconsistencies are due to a lack of standardization of the PRP preparation method, but they do not demonstrate with data whether the quality of the administered PRP is responsible for these treatment failures. Certainly, the great expansion of PRP therapy among medical disciplines has led to the development of several dozen commercial closed systems and various semi-closed methods for obtaining PRP, guided by many other protocols for preparation [14,15]. The final concentrated product is highly susceptible to the centrifugation conditions and the physiognomy of the device, making different systems result in strong disparities in the concentration and cellular and biochemical composition of PRP [14,15,19,27,28].

A recent study by our group revealed differences in the cell density measured in the PRP prepared from the same blood samples of patients with different systems and methods based on blood collection tubes. Moreover, our work warned that the platelet composition and leukocyte differential (density and cell type) are highly dependent on the cell-capturing properties of the device. The new commercial Easy kit system (Mesotech, Napoli, Italy) doubled the content of platelets and leukocytes in PRP preparations compared with other systems commonly used in the clinical routine [29].

The lack of knowledge about the cell capture properties of the selected systems means that the managed PRP is not optimized for clinical purposes; for instance, suboptimal concentrations of platelets and growth factors are used, or leukocytes are randomly introduced or excluded, when their co-administration should be guided by the pathology being treated [19,20].

However, there are other potential factors involved in this controversy. The strong intra-subject [15,30] and, more importantly, inter-subject variabilities modify the quality of the obtained PRP. Variations between subjects are more complex and often related to natural aging, sex, or the presence of underlying diseases [16,31,32,33]. It is widely known that aging has a negative influence on the response to healing and the degeneration of elastic tissues [34]. Protective molecular markers decrease with age [31,32] and more markedly in men than in women [35], highlighting a relationship with hormonal dimorphism: via differential regulation of the expression of multiple growth factors, and the occurrence of these events [36,37]. Estrogens slow the effects of aging, which becomes more noticeable after menopause [37]. This hormonal dimorphism also entails an immunological dimorphism [33,38,39] associated with the differential expression of cytokine receptors [40] and a different predisposition by gender to diseases [40,41].

The purpose of this study was to assess the entire cellular composition of PRP and the influence of sexual dimorphism on the variability of the final product obtained using a new commercial blood separation–concentration system. As a secondary objective, we set out to examine the influence of aging in both sexes on the quality of the concentrated final product.

## 2. Materials and Methods

### 2.1. Study Volunteers

A total of 26 healthy adults, 13 male and 13 female volunteers, with a wide age range, who were not under analgesic drug treatment, were recruited for this study. Anemic subjects or pregnant women were not enrolled. Age, height, and weight data from each subject were collected.

The sample size was calculated using the G*Power 3.1.9.2 software (Heinrich-Heine Universität Düsseldorf, Düsseldorf, Germany) applying the Mann–Whitney U test with a Laplace distribution to detect a large effect size (0.85) in blood counts between males and females with 80% statistical power (α = 0.05, one-tailed hypothesis). A total sample size of 24 participants was calculated with at least 12 participants per group. Our inclusion of 13 participants surpasses the minimum sample size requirement.

Secondly, volunteers of each gender were divided into a “young” participant group, including subjects ≤ 50 years old, and an “older” participant group, with subjects > 50 years old, defining pre- and post-menopausal periods for further characterization of age-related hormonal dimorphism.

### 2.2. Study Procedures

Participation involved 15 mL of blood draw per volunteer. Whole blood (WB) samples were obtained by venipuncture from the cephalic, basilic, or median vein, using a standard aseptic technique. A 4 mL volume of WB was collected into an EDTA tube to carry out the cell blood count of platelets and leukocytes (WBCs) at baseline, including neutrophils, lymphocytes, and monocytes, using a Coulter LH 750 automated analyzer (Beckman Coulter Inc., Brea, CA, USA). The remaining blood sample was used in preparing the PRP by the Easy PRP fractionation–separation blood system (Mesotech, Napoli, Italy). The procedure was developed following the instructions of the manufacturer. Briefly, the syringe containing 1 mL of anticoagulant and 11 mL of WB from each volunteer was gently inverted ten times to evenly mix the components. The blood was incorporated into the lower compartment of the device, through the narrowing of the kit, where after the saturation, the blood rose to fill half of the central body. Then, the device was centrifuged at 3500 rpm for 5 min, and the buffy coat and the fraction of the platelet-rich plasma were extracted using a 5 mL syringe and a 19 G and 90 mm needle. A total of 2 mL of the final product was transferred to an EDTA tube to determine the count of platelets and WBCs in the concentrates.

Platelets and WBCs in WB and in the final concentrated product (PRP) were expressed per milliliter of sample. The increase in the platelet or WBC concentration factor (the mean cell fold change from the WB basal concentration) was calculated by dividing the final concentration of platelets or WBCs in PRP by the initial concentration of platelets or WBCs in WB (PRP/WB ratio) [14]. The PRP/WB ratio was calculated independently for each enrolled volunteer. The results were expressed as the mean and median of the examined population; after, all volunteers (*n* = 26) and the resulting groups were split by sex, or by sex and age.

The PRP/WB ratio vs. the density of each blood cell in WB was modeled by a linear regression using the data of all participants (GraphPad Prism v.8.0, GraphPad Software, San Diego, CA, USA, EE. UU., www.graphpad.com accessed on 4 February 2022).

The CL ratio (CLR; neutrophils, lymphocytes, or monocytes/WBCs), calculated by dividing the concentration of each leukocyte cell type by the WBC concentration in WB, was also related to the PRP/WB ratio by a linear regression.

The results of WB, PRP, and the increase in the cell concentration factor between males and women were compared to establish the influence of hormonal dimorphism by gender and between age–sex groups for age-related hormonal dimorphism.

### 2.3. Statistical Analysis

Continuous variables were reported using the mean, standard deviation (SD), and range, as well as the median and interquartile range (IQR). A paired sample Student *t*-test (*t*-test) or the Wilcoxon test was performed depending on the distribution of the data based on the Shapiro–Wilk test, for comparisons between WB and concentrates. An independent *t*-test or the Mann–Whitney U test (U test) was used for comparisons between male and female groups and between age–sex groups. For all analyses, a value of *p* < 0.05 was considered statistically significant. The data obtained were analyzed using SPSS software for Mac (Version 22; IBM Corp., Armonk, NY, USA).

## 3. Results

The population enrolled had a mean age (±SD) of 62.32 ± 23.43 (29–93) years (median, 70 years) and a body mass index of 25.04 ± 5.13 (17.75–34.58) kg/m^2^ (median, 24.68).

Platelet and WBC counts (mean ± SD; median [IQR]) in WB were 222.10 ± 79.76; 201.00 [122.50] and 7.10 ± 2.76; 6.79 [2.85] cell/mL (see Appendix A). In PRP, platelet and WBC counts increased significantly in all volunteers. The increase in the platelet concentration factor (PRP/WB ratio) was 3.98 times, and the increase in the WBC concentration factor was 1.98 times (neutrophils, ×1.90, lymphocytes, ×2.80, and monocytes, ×2.80 times WB levels). The inter-subject variability in the baseline counts was wide and increased in the PRPs, indicating that the system influenced the final variability of the concentrates (see Appendix A).

Descriptive data of sample volunteers by sex are summarized in Appendix A. All variables, except age, followed a Gaussian distribution (*p* > 0.05; Shapiro–Wilk). Among demographic variables, only differences in height were detected between men and women.

Table 1 shows the density of platelets and WBCs in WB and in PRP in men and women. Strong intra-sex group variability in the WB count was observed. Comparing both groups, platelet and total WBC counts were higher in men than in women, but differences were not statistically significant. However, the WBC differential count was gender-dependent; the lymphocyte count was lower in men (*p* > 0.05), and that of phagocytic cells was significantly lower in women (*p* = 0.035 for neutrophils and *p* = 0.042 for monocytes).

The PRP/WB ratios for platelets and WBCs (Figure 1) also varied markedly among volunteers of each gender and were more pronounced for platelets and the cell fraction in WBCs and in the women group. As shown in Figure 1, the PRP/WB ratios of platelets (*p* = 0.818, *t*-test), neutrophils (*p* = 0.681, U test), lymphocytes (*p* = 0.132, U test), and monocytes (*p* = 0.293, *t*-test) were not different between both genders. However, the increase in the concentration factor (PRP/WB ratio) was higher for platelets (mean, 3.83 and 3.93 times for men and women, respectively) and lymphocytes (mean, 3.03 and 2.61 times, respectively) than for phagocytic cells (mean, 1.97 and 1.92 times for neutrophils; 1.96 and 2.52 times for monocytes, respectively) and specific according to gender; the PRP/WB ratios of platelets, neutrophils, and monocytes were greater in women, and that for lymphocytes was greater in men. This indicates that the efficiency of the device was dependent on the type of blood cell and more favorable in the group with the lowest density of a particular blood cell type (see WB data in Table 1 and Figure 1). In fact, as we detail in Appendix A, there is a relationship between the capture efficiency of the device and the density in WB among subjects, and also between capture efficiency and CLR, in which the capture of cells improves as the density drops in WB and with the decrease in CLR. This trend seemed more pronounced among the different WBCs than platelets for the corresponding range of capture efficiency of each cell type.

Platelet and WBC counts in PRP were significantly higher than in peripheral blood in males (*p* ≤ 0.01) and females (*p* < 0.008) (Table 1). Counts per milliliter of platelets and total leukocytes were not significantly different between males and females (Table 1). WB differences in neutrophils (*p* = 0.035) persisted in the final products (*p* = 0.018). In contrast, the density of monocytes in the final products was not statistically different between men and women (*p* = 0.947).

Table 2 shows the WB and PRP results for men and women after splitting the population by the cut-off age. No significant intra- or inter-gender differences in peripheral blood platelet concentrations were found. In contrast, pronounced differences were observed for WBCs between groups (young men or women vs. older women) and in the leukocyte composition; phagocytic cells were significantly lower in women > 50 years old than in the other groups (with a lower neutrophil count in younger men than younger women > older men, and with a lower monocyte count in older men than younger men > younger women), and lymphocytes were significantly higher in the < 50 year age groups, both within (vs. older volunteers) and between genders (younger men or women vs. older women or men), with the lowest densities in the group of older men.

The PRP/WB ratios in age–sex groups (Figure 2) were virtually identical to those observed for each gender, with a strong decrease in cell capture variability, which was concentered in the monocyte cell line for >50-year-old women. The platelet capture efficiency was similar between the different groups, while the WBC capture efficiency was more favorable in the groups with a lower density of circulating WBCs in WB. For neutrophils, it was more optimal in young men and older women and less optimal in young men; for monocytes, it was more optimal in women > 50 years and less optimal in young men; for lymphocytes, it was more optimal in older men and less optimal in women > 50 years old.

In the final products (see PRP product in Table 2), pre-existing differences between age–sex groups in WB phagocytic cell densities were reduced or even reverted, being significant only for neutrophils between young men and older women. In contrast, for lymphocytes, the significant differences in WB between age–sex groups persisted in the final products.

Overall, WBCs in PRP products were significantly higher in young men (as in WB) and in older men vs. older women, but not among women, as was seen in WB.

## 4. Discussion

In this study, we demonstrated that significant gender differences existed in the leukocyte composition of platelet-rich plasma prepared with commercial cell blood separation devices. These changes were closely related to the sexual immunological dimorphism and age linked to the sex under the strong influence of the selective efficiency of the system in capturing cells; the efficiency was dependent on both the type of concentrated blood cell and its relative density in circulating blood.

As in previous studies, we did not find that the subject’s gender had an influence on the platelet or total WBC count in the prepared PRP final products [16,31,32]. These interesting works focused on the inconsistencies in the results of PRP clinical trials on the variations in the proteomic content of PRP between men and women and with increasing age of the individual. However, they did not check changes in the cellular composition in age–sex groups, when it is known that the presence of WBCs can alter the levels of growth factors [15,20].

The WBC count in PRP samples has frequently been ignored, and very few studies have examined this fraction in PRP [14,15,19,42], largely because the presence of WBCs in the final products is dependent on the extraction system and the centrifugation conditions used [14,15,19,27]. The systematic analysis of the composition of white cells in the concentrated leukocytes has been less exhaustive [15,42].

The co-administration of leukocytes as an adjuvant to PRP is controversial [15,19,20]. WBCs can induce local inflammation and impede tissue recovery in specific procedures such as intervertebral disc regeneration [8]. Excessive neutrophil-derived proteases have been related to prolonged inflammation that leads to delays in wound healing [43]. On the other hand, leukocytes promote platelet activation and the release of platelet-derived growth factors and vascular endothelial growth factors released by leukocytes [15,19,20] and provide the antimicrobial and immunomodulatory capacity to PRP [20,43,44]. In this respect, several studies have reported that leukocytes in PRP improve healing rates of diabetic chronic wounds [9,10]. These wounds are recalcitrant to healing due to vascular and neuropathic effects of diabetes and the inhibitory effects of hyperglycemia on the neutrophil function that impair the immune response, making them unable to control wound infections [45]. In this clinical setting where the patient is at risk and antimicrobial therapy is ineffective [46], co-administration of leukocytes could be interesting to reduce the incidence of amputations.

Regardless of specific interpretations, these studies demonstrate that WBCs have a strong influence on the quality of PRP, and that the final composition of PRP should be guided by the pathology to treat.

In particular, here, we detected significant differences in the neutrophil concentration in the PRP between men and women, which were restricted to specific age–sex groups after splitting each gender by age; young men had significantly higher neutrophils in PRP than women 50 years old or more. The lymphocyte composition also varied significantly between sex–age groups, with age being a greater relevant factor than sex (see Table 2), despite the fact that gender differences were negligible before stratifying the population by age.

Therefore, the gender of the volunteer would be a useful demographic variable to predict differences in the neutrophil content in PRP, but not to predict the leukocyte composition, which was strongly influenced by age.

Differences in the leukocyte composition between age–sex groups were more pronounced in WB than in PRP. The influence of sexual dimorphism and natural aging in the composition of leukocytes in peripheral blood has been described in detail in large donor populations [33,38]. Mononuclear cells of the immune system decrease with aging, although in a more sustained way in women than in men [38], while the density of WB neutrophils significantly increases in men and decreases in women as age advances, with higher concentrations in “young” women < 50 years old than in “young” men [33,39]. This immunological dimorphism is closely related to sex hormones, since delayed neutrophil apoptosis and decreased lymphocyte production are associated with the circulating level of estradiol [33,47] and also with inherent changes in the aging process that result in elevated levels of basal inflammation and an impaired ability to mount efficient innate and adaptive immune responses to pathogens [40,48]. Together, this would explain the evolution in the circulating number of neutrophils and lymphocytes observed in our work in the age–sex groups, as well as the strong decrease in neutrophils and lymphocytes observed in women > 50 years old vs. the young groups, or the strong decrease in lymphocytes, which was greater in men than women, in individuals greater than 50 years old.

A strong limitation of this part of the study was the small sample size available after stratifying male and female volunteers according to the critical age of 50 years. However, despite the sample size, the effects of sex and age previously reported in large series of volunteers [33,38,39,47] were clearly observed in our subject population. Some previously unidentified differences between men and women (lymphocytes) even became tangible after stratification of the population by age.

The decrease in inter-subject variability for all cell fractions in WB and the decrease in the variability of the system efficiency, when we split the population into age groups, reflected this influence of age on sexual dimorphism.

In this way, one might think that the density of circulating WBCs in WB, according to sex–age groups, should be a good predictor of the cell concentration in PRP. However, as we have seen, cell density discrepancies between groups in WB did not correlate with those found in PRP. This was attributed to the different efficiency of the system in capturing cells between genders and between age–sex groups.

First, we found that the system works according to a cell capture gradient in which the smaller cells would be recovered more efficiently, at least using our differential centrifugation protocol. There are many variables (the number of spins, time of centrifugation, centrifugation force, the volume of WB, and the idiosyncrasy of the device itself) in the preparation of PRP that causes marked differences in the composition and cell density in the final product [14,15,27,28]. Piao et al. demonstrated that the working conditions (centrifugation force and time) of the concentrator device that allow the maximum recovery rate for platelets and white blood cells from circulating blood are markedly different. Low-speed centrifugation would improve the capture efficiency of WBCs compared to platelets, due to the higher cell densities and sizes [27]. Our system working conditions follow those recommended by the manufacturer and naturally favor the capture of platelets, to the detriment of the leukocyte fraction (Figure 1). However, based on the same principles that favor platelet capture, the different sizes and densities of the WBC differential also influence the capture efficiency of the system for each type of leukocyte [42]. As we show, the efficiency in the concentration of lymphocytes was practically double that of phagocytic cells, and even quadruple that of other systems. This leads to an over-representation of the lymphocytes in PRP compared to phagocytic cells, with respect to values in peripheral blood, and explains the great inter-subject variation in lymphocyte capture, since lymphocytes comprise a heterogeneous set of cells, varying according to gender and with aging [40]. It is important to stress that the role of lymphocytes would be negligible compared to phagocytic cells in acute infections, if leukocyte infiltration therapy is desired.

Secondly, an efficiency gradient dependent on the particular density of each blood cell in WB was observed; the lower the density in circulating blood, the greater the capture of cells. This system efficiency dependent on the WB density was more pronounced for leukocyte recovery, given the unfavorable acceleration conditions imposed for WBCs compared to platelets.

The sum of effects of both efficiency gradients should be more remarkable for cell types worse captured by the system and poorly concentrated in circulating blood. Both gradients would explain the final composition of the PRP between different age–sex groups. They would also explain that the differences in the density of circulating WBCs (total) between the different age–sex groups do not always translate into differences in the PRP and vice versa. Thus, without taking into account the effects of the system on the different fractions of white cells, the density of WBCs in WB does not predict the density in the concentrated products.

This “system compensatory efficiency” tended to reduce and even reverse the different compositions of WBCs between groups predicted by the circulating blood count (such as the monocyte comparison between genders or several WBC comparisons between age–sex groups). In PRP, only the most marked differences in WB for the cell types with a low capture efficiency persisted significantly, while for cells with a more optimal capture efficiency, these remained unchanged (lymphocytes).

From our point of view, the inter-subject variability, together with the system efficiency variability, should be more important than the intra-subject variability seen after sequential preparations of PRP spaced in time [16]. The latter can be controlled with the result of the patient’s blood count prior to obtaining the PRP and with the skill of the clinician and the accumulated experience in the use of selected systems. In this regard, in our study, the increase in the platelet concentration factor with the Easy-Kit system (≈4.00 × WB platelet count in men and women) was far from the efficiency shown by the manufacturer (7–9 times, https://www.mesotech.it/products) (accessed on 4 February 2022). However, we achieved high repeatability in the capture efficiency among the 26 patients studied for this cell fraction, which seems less affected by demographic variables.

In conclusion, age seems to be a more determining factor than sex in the preparation of PRP, although sexual dimorphism, especially after menopause, influences the composition of leukocytes captured by the system. Thus, age and sex would be demographic factors with a great influence on the quality of PRP both at the proteomic [16] and cellular levels.

In agreement with Xiong et al., a personalized approach to PRP treatment is needed. However, the therapeutic customization should not be limited to the proteomic composition of the PRP; it should be extended to the cellular content, given potential repercussions or benefits, direct or indirect, of the co-administration of leukocytes in the clinical outcome. The sexual immunological dimorphism depending on the age of the subjects can seriously affect our therapeutic purposes when our intention is to use WBCs for their beneficial effects. Age and gender should be taken into account as important modifiers of the composition of PRP, which should be predicted from the cell densities in circulating blood and after knowledge of the capacity of the system to concentrate blood cells and its compensatory efficiency depending on the type of concentrated cell.

Finally, our study is not intended to promote a standard PRP system for purely commercial purposes. It would be impossible to compare all currently available blood separation methods. This concept must be considered regardless of the commercial system or protocol used. For this reason, we propose, as a benchmark, to establish a standardized protocol that will allow an optimized concentration of the different types of blood cells, guiding indications for prepared PRP and warning about the limitations of the different blood concentration methods to fulfill these purposes.

## Figures and Tables

**Figure 1 jcm-11-01683-f001:**
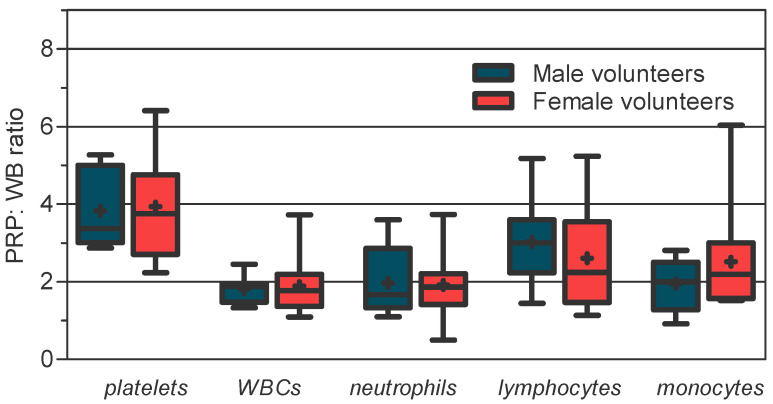
Increase in blood cell concentration factor (PRP/WB ratio) in male and female groups using the Easy PRP kit. Box plot: median, line in the middle of the IQR box; mean, symbol +; whiskers, minimum to maximum. Mean (±SD) and median [IQR] for each blood cell: For male volunteers: platelets, 3.83 (±0.95) and 3.37 [1.99]; WBCs, 1.82 (±0.31) and 1.89 [0.48]; neutrophils, 1.97 (±0.93) and 1.67 [1.53]; lymphocytes, 3.03 (±1.06) and 3.00 [1.37]; monocytes, 1.96 (±0.66) and 2.00 [1.22]. For female volunteers: platelets, 3.93 (±1.22) and 3.75 [2.06]; WBCs, 1.89 (±0.71) and 1.77 [0.82]; neutrophils, 1.92 (±0.91) and 1.86 [0.80]; lymphocytes, 2.61 (±1.30) and 2.24 [2.08]; monocytes, 2.52 (±1.20) and 2.19 [1.43].

**Figure 2 jcm-11-01683-f002:**
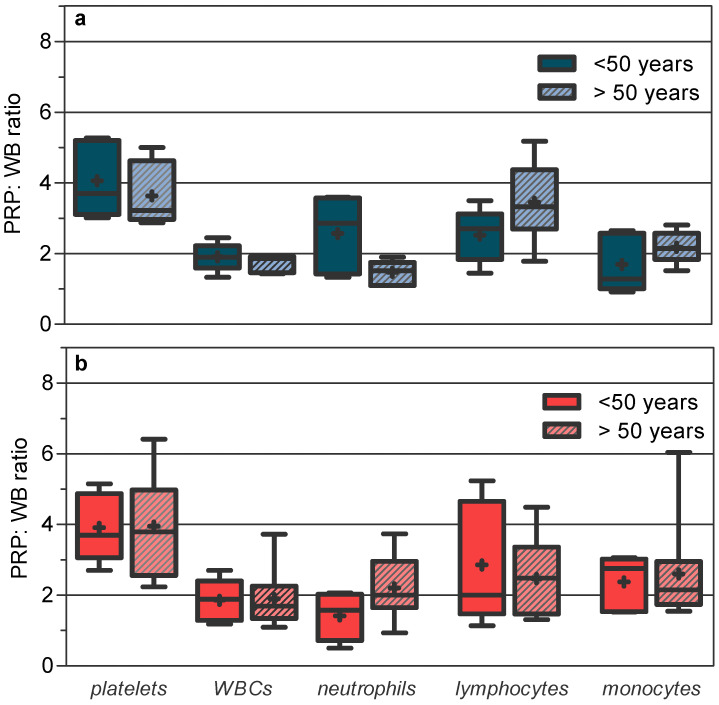
Increase in blood cell concentration factor (PRP/WB ratio) in age–sex groups using the Easy PRP kit: (**a**) men; (**b**) women. Box plot: median, line in the middle of the IQR box; mean, symbol +; whiskers, minimum to maximum. Mean (±SD) and median [IQR] for each blood cell: For the < 50-year-old male group: platelets, 4.06 ± (1.07) and 3.70 [2.10]; WBCs, 1.91 (±0.40) and 1.90 [0.64]; neutrophils, 2.57 (±1.09) and 2.86 [2.16]; lymphocytes, 2.52 (±0.76) and 2.70 [1.29]; monocytes, 1.69 (±0.82) and 1.28 [1.57]. For the >50-year-old male group: platelets, 3.64 (±0.89) and 3.22 [1.66]; WBCs, 1.74 (±0.26) and 1.86 [0.45]; neutrophils, 1.47 (±0.34) and 1.51 [0.65]; lymphocytes, 3.45 (±1.15) and 3.32 [1.68]; monocytes, 2.18 (±0.46) and 2.15 [0.74]. For the female < 50-year-old volunteers: platelets, 3.91 (±0.97) and 3.69 [1.82]; WBCs, 1.86 (±0.60) and 1.89 [1.12]; neutrophils, 1.41 (±0.68) and 1.57 [1.32]; lymphocytes, 2.85 (±1.72) and 2.00 [3.18]; monocytes 2.37 (±0.77) and 2.76 [1.49]. For the female >50-year-old volunteers: platelets, 3.95 (±1.39) and 3.79 [2.43]; WBCs, 1.90 (±0.80) and 1.69 [0.92]; neutrophils, 2.20 (±0.93) and 2.00 [1.32]; lymphocytes, 2.47 (±1.09) and 2.48 [1.89]; monocytes, 2.60 (±1.42) and 2.15 [1.2].

**Table 1 jcm-11-01683-t001:** Platelet and WBC counts (cells × 10^6^/mL) in WB and in the final PRP product prepared by using the Easy PRP Kit in male and female volunteers.

		Male Group *n* = 13			Female Group *n* = 13			*p* Value
Blood Cell Line		Mean ± SD (Range)	Median (IQR)	SW *	Mean ± SD (Range)	Median (IQR)	SW *	
Platelets	WB	252.00 ± 81.88 (161.00–382.00)	228.00 [158.00]	0.081	198.60 ± 72.39 (56.00–334.00)	188.00 [104.80]	0.731	0.097 ^b^
	PRP product	926.00 ± 222.40 (462.70–1222.00)	945.00 [337.50]	0.760	757.10 ± 329.60 (359.00–1317.00)	674.90 [602.80]	0.177	0.159 ^b^
WBCs	WB	7.99 ± 1.97 (1.02–11.70)	7.53 [3.09]	0.466	7.06 ± 2.88 (3.90–14.80)	6.35 [3.02]	0.029	0.171 ^a^
	PRP product	14.55 ± 4.72 (4.59–24.00)	13.80 [5.43]	0.092	12.67 ± 5.29 (7.08–23.64)	9.48 [9.15]	0.036	0.180 ^a^
Neutrophils	WB	4.83 ± 1.66 (2.26–8.70)	4.93 [3.03]	0.737	3.47 ± 1.57 (1.81–6.59)	2.81 [2.30]	0.025	0.035 ^a^
	PRP product	8.33 ± 1.60 (4.08–16.99)	8.12 [3.46]	0.313	5.98 ± 2.70 (2.36–11.70)	5.51 [3.46]	0.277	0.018 ^b^
Lymphocytes	WB	1.94 ± 0.85 (0.77–3.06)	1.93 [1.66]	0.196	2.19 ± 1.15 (0.72–5.45)	2.20 [1.27]	0.026	0.848 ^a^
	PRP product	5.23 ± 1.68 (3.67–8.26)	4.88 [2.79]	0.014	4.99 ± 2.48 (2.23–11.98)	4.34 [2.89]	0.014	0.460 ^a^
Monocytes	WB	0.58 ± 0.17 (0.31–0.84)	0.57 [0.27]	0.721	0.45 ± 0.15 (0.20–0.73)	0.45 [0.25]	0.956	0.042 ^b^
	PRP product	1.07 ± 0.31 (0.67–1.60)	0.97 [0.63]	0.351	1.08 ± 0.50 (0.41–2.24)	0.91 [0.79]	0.207	0.947 ^b^

Abbreviations: WBCs, leukocytes; SD, standard deviation; range (min–max); IQR, interquartile range; * SW, Shapiro–Wilk test. ^a^ Mann–Whitney U test; ^b^ Independent *t*-test for comparisons between male and female groups. A value of *p* < 0.05 (with a 95% confidence interval) was considered statistically significant.

**Table 2 jcm-11-01683-t002:** Blood cell counts (mean cells ×10^6^/mL ± SD; median [IQR]) in WB and in the final product in male and female volunteers according to age-based breakpoint.

	Male Volunteers				Female Volunteers			
Blood Cell Line	≤50 Years (*n* = 5) ^a^		>50 Years (*n* = 8) ^b^		≤50 Years (*n* = 5) ^c^		>50 Years (*n* = 8) ^d^	
	WB	PRP Product	WB	PRP Product	WB	PRP Product	WB	PRP Product
Platelets	269.40 ± 97.17 228.00 [186.50]	1026.00 ± 197.10 1096.00 [374.20]	237.50 ± 72.79 212.00 [132.80]	842.40 ± 222.30 911.30 [383.80]	236.40 ± 72.73 207 [133.50]	905.10 ± 292.40 902.40 [540.50]	177.70 ± 66.96 173.00 [77.50]	674.80 ± 335.30 624.00 [534.00]
WBCs	8.44 ± 2.36 ^&^ 8.10 [4.46]	16.43 ±6.68 ^&^ 16.20 [13.24]	7.62 ± 1.73 7.52 [3.38]	13.00 ± 1.54 ^+^ 13.42 [3.01]	9.03 ± 3.44 * 7.60 [5.52]	15.86 ± 4.42 17.40 [7.68]	5.96 ± 1.95 5.60 [2.33]	10.90 ± 5.08 9.48 [3.49]
Neutrophils	4.45 ± 2.07 ^&^ 3.95 [4.00]	9.64 ± 1.14 ^&&,^** 10.14 [2.12]	5.14 ± 1.35 ^+++^ 5.04 [2.35]	7.24 ± 0.97 6.86 [1.80]	5.27 ± 1.15 ** 4.90 [2.23]	6.91 ± 2.68 6.24 [4.97]	2.47 ± 0.43 2.40 [0.65]	5.46 ± 2.72 4.39 [2.84]
Lymphocytes	2.56 ± 0.48 *^,&^ 2.70 [0.93]	6.37 ± 1.93 * 6.75 [3.76]	1.42 ± 0.73 ^&^ 1.25 [0.80]	4.29 ± 0.56 ^&^ 4.08 [1.04]	3.01 ± 1.43 * 2.50 [2.19]	7.23 ± 2.84 ** 6.23 [4.40]	1.73 ± 0.70 1.70 [1.19]	3.74 ± 1.04 3.64 [1.34]
Monocytes	0.63 ± 0.22 ^&^ 0.61 [0.40]	0.96 ± 0.34 0.78 [0.57]	0.54 ± 0.11 ^+^ 0.53 [0.18]	1.17 ± 0.28 1.06 [0.51]	0.58 ± 0.11 ** 0.61 [0.22]	1.32 ± 0.34 1.24 [0.60]	0.37 ± 0.11 0.35 [0.17]	0.95 ± 0.54 0.76 [0.64]

Abbreviations: WBC, leukocytes; SD, standard deviation; IQR, interquartile range. Statistical analysis was performed according to the normality test reported in Table 2. ^a^ Mean age (±SD) and median age [IQR]: 38.40 (±6.84) and 38.00 [13.00]. ^b^ Mean age (±SD) and median age [IQR]: 81.83 (±10.68) and 84.50 [19.00]. ^c^ Mean age (±SD) and median age [IQR]: 34.00 (±2.55) and 34.00 [4.00]. ^d^ Mean age (±SD) and median age [IQR]: 78.33 (±10.77) and 75.00 [20.00]. * *p* < 0.05, ** *p* < 0.01 for WB or PRP product between age groups in men or in women. ^+^
*p* < 0.05, ^+++^
*p* < 0.001 for WB or PRP products between male and female volunteers <50 or >50 years old. ^&^
*p* < 0.05, ^&&^
*p* < 0.01 for WB or PRP products between male < 50 years and female > 50 years groups or between male > 50 years and female < 50 years groups.

## Data Availability

The data presented in this study are available on request from the corresponding author.

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
