# Peer review of "Influence of Sexual Dimorphism, Aging, and Differential Cell Capture Efficiency of Blood Separation Systems on the Quality of Platelet-Rich Plasma"

_jcm, 2022, doi:10.3390/jcm11061683_

Round 1

Reviewer 1 Report

In this manuscript, the author checked the impact of hormonal/immunological dimorphism of patients in the cellular composition of platelet-rich plasma products-(PRP). The platelet and white blood cells-(WBCs) count in WB and PRP from 26 volunteers were compared between genres and after split population into pre-(≤50 years) and post-(>50 years) menopausal ages. The study in the paper could provide a benchmark to establish a standardized protocol that will allow an optimized concentration of the different types of blood cells, guiding indications for prepared PRP and warning about the limitations of the other blood concentration methods to fulfill these purposes. However, there are some issues to be addressed before further review on the result of this manuscript, and I would like to address several points as follows:

Major comments:

  1. In this paper, the platelet and white blood cells-(WBCs) count in WB and PRP were compared between genres and after split population into pre-(≤50 years) and post-(>50 years) menopausal ages. However, the lack of collected cases of clinical volunteers is a drawback since one of the studies aims to compare currently available blood separation methods.

  1. In line 169 to 173, “the increase in the concentration factor was higher for platelets (mean, 3.83- and 3.93-times for men and women, respectively) and lymphocytes (mean, 3.03- and 2.61-times, respectively) than for phagocytic cells (mean, 1.97- and 1.92-times for neutrophils; 1.96- and 2.52-times for monocytes, respectively) and specific according to gender”. It seems that “83- and 3.93-times for men and women”, “3.03- and 2.61-times”, and “1.97- and 1.92-times for neutrophils; 1.96- and 2.52-times for monocytes” could not be calculated by the data in Table 1.

Besides, as shown in Table 1, the lymphocytes in WB of male group are higher than the female group. However, the lymphocytes in PRP product showed an opposite trend, not like the trend of platelets, WBC, neutrophils, and monocytes in WB and PRP product.

  1. In line 196 to 197, “In contrast, the density of monocytes was not statistically different between men and women” is not consistent with the statistical result of monocytes (P=0.042) in Table 1.

In Figure 1, the legends or notes is difficult to understand, “Platelet; 3.83 ± 0.95; 3.37 [1.99], WBC; 1.82 ± 0.31; 1.89 [0.48], neutrophils; 1.97 ± 0.93; 1.67 [1.53], lymphocytes; 3.03 ± 1.06; 3.00 [1.37] and monocytes; 1.96 ± 0.66; 2.00 [1.22] for male volunteers”. It seems that the punctuation is messy, “; or ,”. Besides, what does the asterisk mean in Figure 1? Which groups are statistically compared?

Similar problems also exist in other charts.

  1. In Table 2, the blood cell counts (mean cells x106 /ml ± SD; median [IQR]) in WB and the final product in male and female volunteers were compared between male < 50 years and female > 50 years groups or between male >50 years and female < 50 years groups. However, it is unreasonable to compare the variables of different ages under the same gender or the other gender under different ages.

Minor comments:

  1. In Table 1, line 2, I suppose that “Mean ± DS” should be “Mean ± SD”.

  1. In Table 2, line 2, can you explain the mean of “38.40 ± 6.84; 38.00 [13.00]”, “81.83 ± 10.68; 84.50 [19.00]”?

Author Response

Thank you for your extensive review. Please see the attached file.

Reviewer 2 Report

Thank you for asking me to review this paper which looks at variability in white cell counts in PRP according to age and gender.  The findings are interesting and it is important to note that variability exists.   I have several points to make:

  1. The quoted centrifugation speed for production of PRP seems very high - is it correct?
  2. I do not understand this sentence:

    "Piao y cols, revealed that there exist different 316 critical accelerations and times to achieve the maximum recovery rate for platelets"

  3. The potential clinical relevance of this finding needs further discussion.  Does the difference matter if patients are given autologous PRP for therapeutic uses?  Are there any plans to look at PRP of differing compositions and its associated clinical outcomes?

Round 2

Reviewer 1 Report

The author added necessary information on the data statistics, which greatly improves the manuscript's quality. Besides, the author also made many useful changes in other parts of the manuscript. Only a small question needs to be answered or explained by the author:

  1. In line 180 to 182, “However, the increase in the concentration factor was higher for platelets (mean, 3.83- and 3.93-times for men and women, respectively)” in the new manuscript, can you explain or show the calculation process in detail?

Author Response

Thank you for reveviewing this manuscript. Please see the attached document.
